# ESTAN: Enhanced Small Tumor-Aware Network for Breast Ultrasound Image Segmentation

**DOI:** 10.3390/healthcare10112262

**Published:** 2022-11-11

**Authors:** Bryar Shareef, Aleksandar Vakanski, Phoebe E. Freer, Min Xian

**Affiliations:** 1Department of Computer Science, University of Idaho, Idaho Falls, ID 83402, USA; 2Department of Industrial Technology, University of Idaho, Idaho Falls, ID 83402, USA; 3Department of Radiology and Imaging Sciences, University of Utah School of Medicine, Salt Lake City, UT 84132, USA

**Keywords:** breast ultrasound, tumor segmentation, deep learning, small tumor-aware network

## Abstract

Breast tumor segmentation is a critical task in computer-aided diagnosis (CAD) systems for breast cancer detection because accurate tumor size, shape, and location are important for further tumor quantification and classification. However, segmenting small tumors in ultrasound images is challenging due to the speckle noise, varying tumor shapes and sizes among patients, and the existence of tumor-like image regions. Recently, deep learning-based approaches have achieved great success in biomedical image analysis, but current state-of-the-art approaches achieve poor performance for segmenting small breast tumors. In this paper, we propose a novel deep neural network architecture, namely the Enhanced Small Tumor-Aware Network (ESTAN), to accurately and robustly segment breast tumors. The Enhanced Small Tumor-Aware Network introduces two encoders to extract and fuse image context information at different scales, and utilizes row-column-wise kernels to adapt to the breast anatomy. We compare ESTAN and nine state-of-the-art approaches using seven quantitative metrics on three public breast ultrasound datasets, i.e., BUSIS, Dataset B, and BUSI. The results demonstrate that the proposed approach achieves the best overall performance and outperforms all other approaches on small tumor segmentation. Specifically, the Dice similarity coefficient (DSC) of ESTAN on the three datasets is 0.92, 0.82, and 0.78, respectively; and the DSC of ESTAN on the three datasets of small tumors is 0.89, 0.80, and 0.81, respectively.

## 1. Introduction

Breast ultrasound (BUS) imaging is an effective screening method due to its painless, noninvasive, nonradioactive, and cost-effective nature. Breast ultrasound image segmentation aims to extract tumor region(s) from normal breast tissues in images. It is an essential step in BUS computer-aided diagnosis (CAD) systems. However, because of the speckle noise, poor image quality, and variable tumor shapes and sizes, accurate BUS image segmentation is challenging.

According to the National Cancer Institute, in the United States, the relative survival is 99% if breast cancer is detected and treated at the early stages, and only 27% if cancer has spread to other organs of the body [1]. The early detection of breast tumors is the key to reducing the mortality rate. However, in the early stages, most tumors are small and occupy a relatively small region in BUS images. It is challenging to distinguish them from normal breast tissues. Therefore, the accurate detection of small tumors is critical for early breast cancer detection and can improve clinical decisions, treatment planning, and recovery.

**Table 1 healthcare-10-02262-t001:** Deep learning approaches for BUS image segmentation.

Article	Year	Methods *	Dataset Size	Evaluation Metrics *
Huang et al. [2]	2018	FCN + Wavelet features + CRFs	325	TPR, FPR, JI
Yap et al. [3]	2018	Patch-based LeNet, U-Net, and AlexNet	469	TPR, FPR, F1
Amiri et al. [4]	2020	Transfer Learning	163	DSC
Nair et al. [5]	2020	Deep Neural Networks + Two Decoders + Simulated Data	22230	DSC
Zhuang et al. [6]	2019	U-Net + Attention gate	1062	TPR, Sp, F1, Pr, JI, Acc, DSC, AUC
Hu et al. [7]	2019	Dilated FCN + Active contour model	570	DSC, MAD, and HD
Vakanski et al. [8]	2020	U-Net + Attention blocks	510	TPR, FPR, DSC, JI, Pr, AUC-ROC
Byra et al. [9]	2020	U-Net + Attention gate + Entropy maps	269	DSC, JI
Moon et al. [10]	2020	Ensemble CNNs	246	TPR, FPR
Lee et al. [11]	2020	U-Net + Channel attention module	163	FPR, F1, JI, AUC, Pr, Sp, TPR
Chen et al. [12]	2022	U-Net + Bidirectional attention + refinement residual net	780	Acc, DSC, Sens, Sp, Pr, JI
Hussain et al. [13]	2022	U-Net + level set	349	Acc, DSC, JI
Shareef et al. [14]	2020	U-Net + Two encoders	725	TPR, FPR, JI, DSC, AER, MAE, HD

* TPR: true positive rate, FPR: false positive rate, JI: Jaccard indices, DSC: dice similarity coefficient, Sp: specificity, F1: F_1_ score, Pr: precision, Acc: Accuracy, AUC: area under curve, AER: area error rate, MAD: mean absolute deviation, HD: average Hausdorff distance, ROC: receiver operating characteristic curve, Sens: sensitivity, MAE: mean area error, CRFs: conditional random fields, and FCN: fully convolutional network.

The approaches of BUS image segmentation can be classified into traditional approaches and deep learning-based approaches. Numerous traditional approaches have been used for BUS image segmentation, such as thresholding [15,16,17,18,19,20,21], region growing [22,23], and watershed [24,25]. Despite their simplicity, these methods require knowledge and expertise in extracting features, and they are not robust due to poor scalability and high sensitivity to noise. Refer to [26] for a comprehensive review of BUS image segmentation.

Recently, several deep learning approaches [2,3,4,5,6,7,8,9,10,11,12,13,14] have been developed for BUS image segmentation; Table 1 lists the most recent deep learning approaches for BUS image segmentation. Huang et al. [2] proposed a fuzzy fully convolutional network to perform BUS image segmentation. Fuzzy logic is adopted to solve the uncertainty issue in the BUS images and feature maps. Contrast enhancement and wavelet features were applied as preprocessing techniques to augment the training data. The augmented training image set and features from convolutional layers were transformed into a fuzzy domain by a fuzzy membership function. The context information and the human breast structure were integrated into Conditional Random Fields (CRFs) to enhance the segmentation results. Yap et al. [3] evaluated the performance of three different deep learning approaches: a patch-based LeNet, a U-Net, and transfer learning with a pretrained AlexNet on two BUS datasets (Dataset A and Dataset B). The transfer learning AlexNet outperformed all others on Dataset A for true positive and F-measure metrics and patch-based LeNet achieved the best results on Dataset B for false positive per image metric. Although the results show that the different deep learning approaches designed for other tasks can be adopted and trained on BUS datasets, all the approaches could not achieve the best results for all the evaluation metrics on both datasets. Amiri et al. [4] studied transfer learning and the significance of fine-tuning configurations of U-Net architecture to solve the issue of scarce ultrasound image data. Fine-tuning the shallow layers of U-Net for small BUS datasets achieved the best results; however, there is no significant difference in fine-tuning the whole network or shallow layers for large BUS dataset. Refer to [26,27] for more deep learning approaches for medical image segmentation.

In addition, Nair et al [5] proposed a DNN with two decoders to create BUS images and segmentation masks from raw single-plane wave channel data. This approach showed promising results where both the segmentation masks and B-mode images were generated in a single network using raw data. Zhuang et al. [6] proposed an RDAU-Net model, based on U-Net architecture, to perform the tumor segmentation task on BUS images. The dilated residual blocks and attention gates were used to replace the basic blocks and original skip connections in U-Net, respectively. The RDAU-Net design improves the overall sensitivity and accuracy of the model. Similarly, Hu et al. [7] proposed a DFCN method that combines the dilated fully convolution network with a phase-based active contour (PBAC) model to automatically segment breast tumors. The DFCN with PBAC network is more robust to noise and blurry boundaries, and successfully segments tumors with a large volume of shadows.

Moreover, Vakanski et al. [8] integrated radiologists’ visual attention with a U-Net model to perform BUS segmentation. The model designs attention blocks to ignore regions with low saliency and emphasize more regions with high saliency. This study outperformed the U-Net model, and has successfully combined prior knowledge information into a convolutional neural network. Byra et al. [9] proposed a deep learning segmentation approach for BUS images based on entropy parametric maps with the attention-gated U-Net network. The model achieved a good improvement; however, there are insufficient results and analysis to show the significance of entropy maps. Furthermore, Moon et al. [10] proposed an ensemble CNN architecture for a CAD system comprising multi-models trained on original BUS images, segmented image tumors, tumor masks, and fused images. The fused images were prepared by combining an original image, segmented tumor, and tumor shape information (TSI). The results show that the fused images achieved the best results among all others, and the study provides a clear guide to choosing an approach for a specific dataset size. Lee et al. [11] proposed a channel attention module with multi-scale grid average pooling for segmenting BUS images. The approach utilizes both local and global information and achieves good overall segmentation performance. Chen et al. [12] proposed bidirectional attention and refine network that they added on top of the U-net to accurately segment breast lesions. However, training such a network on a small dataset makes it challenging to deal with overfitting/underfitting issues. These methods achieved good overall performance. However, as shown in Figure 1, they failed to achieve good performance in segmenting small tumors. First, these methods were designed to improve the overall performance using general-purpose square kernels that were developed for learning features in natural images. Second, all currently available BUS datasets are small, and most deep learning-based approaches require a large and high-quality training set.

This work is inspired by the current progress in small object detection and/or segmentation, which is an important task in computer vision, as it forms the foundation of many image-related tasks, such as remote sensing, scene understanding, object tracking, instance and panoptic segmentation, aerospace detection, and image captioning. Chen et al. [28] proposed an augmented technique for the R-CNN algorithm with a context model and small region proposal generator, which was the first benchmark dataset for small object detection. Krishna et al. [29] designed a Faster R-CNN model with a modified upsampling technique to improve the performance of small object detection. Guan et al. [30] proposed a semantic context-aware network (SCAN), which integrates a location fusion module and context fusion module to detect semantic and contextual features. The DenseU-Net architecture was proposed by Dong [31] for the semantic segmentation of small objects in urban remote sensing images. It uses residual connections and a weighted focal loss function with median frequency balancing to improve the performance of small object detection. To the best of our knowledge, STAN [14] was the first deep learning architecture designed specifically for small tumor segmentation. Three skip connections and two encoders were employed to extract multi-scale contextual information from different layers of the contracting part. The Small Tumor-Aware Network outperformed other deep learning approaches for segmenting small tumors in BUS images. However, its average false positive rate (FPR) on small tumors is much larger than the FPR on large tumors.

In this paper, we extend STAN and propose a new architecture, namely the Enhanced Small Tumor-Aware Network or ESTAN, to achieve robust segmentation for tumors of different sizes. The new architecture has two encoder branches. The basic encoder has five blocks and learns features at different scales. The ESTAN encoder applies row-column-wise kernels to adapt to the breast anatomy during feature learning. Specifically, the human breast anatomy consists of four main layers: skin, premammary (subcutaneous fat), mammary, and retromammary layers [32]. Each layer is characterized by a distinct texture and corresponding echo patterns in ultrasound images. The tissue layers in BUS images appear vertically stacked, with similar echo patterns propagating horizontally across images. Breast pathology originates predominantly in the mammary layer. The row-column-wise kernels were designed to learn the breast tissue structure and thus improve detecting small tumor representations in BUS images. In the decoder, each block has three skip connections that fuse rich contextual features from the two encoders. The contextual features are robust to different tumor sizes and help distinguish tumor regions from normal regions.

The rest of the paper is organized as follows: Section 2 presents the proposed architecture; Section 3 demonstrates experimental results and implementation details; and Section 4 and Section 5 are the discussion and conclusion, respectively.

## 2. Enhanced Small Tumor-Aware Network

In this section, we introduce the proposed Enhanced Small Tumor-Aware Network (ESTAN) for solving the issue of small tumor segmentation in BUS images. The Enhanced Small Tumor-Aware Network builds upon two observations: (1) BUS images contain tumors of a broad range of sizes, and current state-of-the-art approaches have poor performance in segmenting small tumors; and (2) the current deep learning-based approaches use square-shape kernels and have difficulty utilizing context information of BUS images, e.g., breast tissue anatomy. To alleviate these challenges, we propose ESTAN to extract and fuse image context information at different scales. The Enhanced Small Tumor-Aware Network constructs feature maps using both square and large row-column-wise kernels. These feature maps extract multi-scale context information and preserve fine-grained tumor location information. Therefore, the new design enables ESTAN to accurately segment breast tumors of different sizes and is especially effective in segmenting small tumors. The overall architecture of the proposed approach is shown in Figure 2.

### 2.1. Basic Encoder

The Enhanced Small Tumor-Aware Network consists of two encoders—the basic and ESTAN encoders. The basic encoder downsamples the input feature maps to extract low-level spatial and contextual information. The basic encoder comprises five blocks, where each of the first four blocks contains two convolutional layers and a max pooling layer, and the fifth block only has two convolutional layers. The basic blocks in the encoder are different from the original U-Net [33] encoder blocks since the new architecture uses two skip connections to copy feature maps from the encoder blocks to the corresponding upsampling layers in the decoder module. Figure 2c illustrates the architecture of the basic encoder.

Let X∈ ℝh×w×c denote the input images, where *h*, *w,* and *c* are the height, width, and number of channels, respectively. Let *f* be the convolution function for square kernels followed by a rectified linear unit (ReLU) activation function, Ki be the number of kernels, and Si be kernel size in the *i*th convolution layer. The output of the *j*th block of the basic encoder is defined by
(1)Bj=ϕ(fS2,K2(fS1,K1(X))) 
where Bj is the output, and ϕ  is the pooling operation. Additionally, the kernel size S1 and S2 in Basic Block 1, 2, 3, 4, and 5 are all set to 3. The number of kernels K1 and K2 in Basic Block 1, 2, 3, 4, and 5 have values 32, 64, 128, 256, and 512, respectively.

### 2.2. ESTAN Encoder

The receptive field in CNNs is important in building effective feature maps that model contextual information. It defines the input image region that impacts output features, and image regions outside the receptive field of a feature will not contribute to the feature calculation. To ensure the coverage of all relevant image regions and achieve enhanced performance, many dense prediction tasks used large receptive fields [34,35]. Several techniques have been applied to increase the receptive field, such as stacking more layers, sub-sampling, and dilated convolutions [36]. However, in BUS images, a large receptive field can result in poor performance for small tumor segmentation [37]. The goal of the ESTAN encoder is to avoid the large receptive field and capture small tumors.

The Small Tumor-Aware Network [14] proposed a two-encoder architecture and applied kernels of sizes 1 × 1, 3 × 3 and 5 × 5. The small kernel size can avoid a large receptive field. The two encoders fused contextual information at different scales by producing features using different sizes of receptive fields. This design improved the overall performance of small breast tumor segmentation. However, STAN produced high false positives for some BUS images with small tumors.

To overcome this problem, we redesigned the encoder by applying row-column-wise kernels. The small square kernels in STAN constructed feature maps using only square image regions. The motivation for the design is because BUS images are composed of vertically stacked tissue layers (Figure 3). Applying row-column-wise kernels in CNNs can avoid calculating features using image regions from multiple anatomical layers and produce more accurate and meaningful feature maps. In addition, in this study, ESTAN is compared with nine state-of-the-art approaches on three datasets, while STAN was compared with only three state-of-the-art approaches on two datasets.

The ESTAN encoder comprises five blocks, named ESTAN blocks, which are parallel with the basic encoder blocks. Each block has four square kernels and two row-column-wise kernels in two parallel branches. Such kernels can efficiently extract contextual and fine-grained details of small tumors in the BUS images. Furthermore, ESTAN blocks add one extra non-linearity to each encoder block. Figure 2b illustrates the design of each ESTAN block. Let $C_{i}$ be the number of kernels, and $A_{i}$ be the kernel size. The output of *j*th ESTAN block is defined by
(2)$$E_{j}=\phi \left(f_{A_{5},C_{5}}\left(f_{A_{2},C_{2}}\left(f_{A_{1},C_{1}}\left(X\right)\right)+f_{A_{4},C_{4}}\left(h_{1,A_{3},C_{3}}\left(h_{A_{3},1,C_{3}}\left(X\right)\right)\right)\right)\right)\;$$
where $E_{j}$ is the output of the jth ESTAN block, and ϕ  is the pooling operation, h is the row-column-wise convolution function followed by a rectified linear unit (ReLU) activation function with the size of $A_{3}\times 1$ and $1\times A_{3}$, respectively. The size of $A_{3}$ in ESTAN Block 1, 2, 3, 4, and 5 are 15, 13, 11, 9, and 7, respectively. The size of $A_{5}$ in ESTAN Block 2 and 5 is 5, and in the rest is 1. Furthermore, block 5 has no pooling operation for both encoders. Moreover, the number of kernels $\left(C_{i}\right)$ in each ESTAN Block 1, 2, 3, 4, and 5 have values 32, 64, 128, 256, and 512, respectively.

### 2.3. Decoder and Skip Connections

The decoder module comprises four upsampling blocks, where each has one upsampling layer followed by three convolution layers. Unlike the U-Net architecture, where the decoder has two convolution layers, the ESTAN adds an additional kernel after the first convolution kernel to control the post-concatenation channels. Let f be the convolution function followed by a rectified linear unit (ReLU) activation function, $Y_{i}$ be the number of kernels, and $M_{i}$ be the kernel size. The output of the jth block of the decoder is defined by
(3)$$U_{j}=f_{M_{3},Y_{3}}\left(f_{M_{2},Y_{2}}\left(f_{M_{1},Y_{1}}\left(\Psi \right)\right)\right)\;$$
where Ψ is the upsampling layer. Kernel sizes $M_{1}$ and $M_{3}$ in all blocks are 3, $M_{2}$ in blocks 1, 2, and 3 is 1, and $M_{2}$ in Block 4 is 5. In addition, $Y_{1},Y_{2}$ and
$Y_{3}$, which represent the numbers of kernels in *j*th Up Block, have the same values in each block, and their values are 256, 128, 64, and 32, respectively.

We have introduced two skipping connections to copy feature maps at different scales from two encoders to the decoder. The first skip connection combines the result of $f_{S_{1},K_{1}}$ in the basic encoder block and the result of $f_{A_{5},C_{5}}$ in the ESTAN encoder block and are concatenated to the upsampling layer. The second skip connection concatenates the results of $f_{S_{2},K_{2}}$ and $f_{M_{2},Y_{2}}$. The output layer utilizes a 1 × 1 convolution followed by a sigmoid activation to predict the final results. Figure 2d illustrates the decoder block.

## 3. Experimental Results

### 3.1. Datasets, Evaluation Metrics and Setup

We use three public BUS datasets: BUSIS [20,26,38,39], BUSI [40] and Dataset B [3]. The BUSIS dataset contains 562 images collected from three hospitals using GE VIVID 7, LOGIQ E9, Hitachi EUB-6500, Philips iU22, and Siemens ACUSON S2000. The BUSIS dataset includes 306 benign and 256 malignant breast ultrasound images. The BUSI dataset is from Baheya Hospital for Early Detection & Treatment of Women’s Cancer in Egypt using the LOGIQ E9 ultrasound system and the LOGIQ E9 Agile ultrasound system with ML6-15-D Matrix linear probe transducers. The BUSI dataset has 780 images, of which there are 133 normal, 487 benign, and 210 malignant images collected from 600 women patients aged 25 to 75 years old. In addition, radiologists from Baheya Hospital reviewed and modified the ground truth masks. The Dataset B has only 163 breast ultrasound images, and the UDIAT Diagnostic Centre of the Parc Taul’ı Corporation, Sabadell (Spain) collected the images using a Siemens ACUSON Sequoia C512 system with a 17L5 linear array transducer (8.5 MHz). Dataset B consists of 53 malignant, and 110 benign images from different women with a mean image size of 760 × 570 pixels. The Dice loss [41] function is used in this work.

The tumor size is an important variable, and Figure 4 illustrates the histograms of tumor size distributions of the three datasets based on their original resolution. The physical sizes of most tumors in the three datasets are unavailable; therefore, we define the tumor size as the length (in pixels) of the longest axis of a tumor region in the original BUS image. The distributions of BUSI and Dataset B show positive skewness where many tumors are smaller than 150 pixels. The BUSI dataset has more large tumors compared to the other datasets, and the sizes of most tumors are between 150 and 250 pixels. In addition, the images in the BUSIS dataset were collected with five different BUS workstations; thus, the image quality has large variations.

To evaluate the segmentation results, both area and boundary metrics are employed. The metrics are true positive rate (TPR), false positive rate (FPR), Jaccard index (JI), Dice similarity coefficient (DSC), area error rate (AER), Hausdorff distance (HD), and mean absolute error (MAE). For detailed information about the seven metrics, refer to [26]. We perform five-fold cross-validation individually for each dataset to evaluate the test performance of all methods, and the input image size is 256 × 256 pixels for all the approaches. In this study, we compare the proposed method with nine state-of-the-art approaches: AlexNet [42], SegNet [37], U-Net [33], CE-Net [43], MultiResUNet [44], RDAU-Net [6], SCAN [30], DenseU-Net [31], and STAN [14]. These approaches have different backbone networks and different training strategies. We employ a transfer learning technique for AlexNet, which is pretrained on ImageNet. SegNet, U-Net, CE-Net, MultiResUNet, RDAU-Net, SCAN, and DenseU-Net are trained from scratch.

Note that the FPR is calculated as the ratio between the number of false positives and the total number of actual positives [22,26,38], which is different from the commonly used FPR formulation as the ratio between the number of false positives and actual negatives. In the definition, if the size of false positive regions is larger than the size of the actual positive regions, FPR will be greater than 1. The new FPR definition is preferred in BUS image segmentation because of the large size of negative regions (denominator) in the old FPR.

All experiments are performed on a workstation with a 3.50 GHz Intel(R) Xeon(R) CPU, 32 GB of RAM, and an Nvidia Titan Xp GPU.

### 3.2. Overall Performance

In this section, we compare the proposed approach with AlexNet, SegNet, U-Net, CE-Net, MultiResUNet, RDAU-Net, SCAN, DenseU-Net, and STAN. The results are shown in Figure 5 and Table 2.

Figure 5 shows the segmentation results of four sample BUS images. In the first row, the tumor in the BUS image is small, and AlexNet, U-Net, MultiResUNet, SCAN, and DenseU-Net have poor segmentation performance. In the second and third samples (second and third rows), all approaches, except the proposed ESTAN, produce a high false positive, which demonstrates that they have difficulty distinguishing tumor regions from tumor-like regions. In Figure 5k, STAN can segment small tumors accurately but still produces false tumor regions. Figure 5l shows that ESTAN segments the four images accurately without any false tumor regions.

Table 2 presents the quantitative results of all approaches on the three datasets. The proposed ESTAN achieved the best overall performance on all three datasets. AlexNet and SegNet obtained high TPRs, but at the cost of high FPRs.

To investigate the statistical significance of all approaches, the Wilcoxon signed-rank test was employed to compare ESTAN against all other approaches for FPR, JI, DSC, AER, HE, and MAE metrics on the three datasets. The significance level is defined as *p*-value < 0.05. The obtained *p*-values from the Wilcoxon signed-rank test were corrected using the Holm–Bonferroni method for multiple comparisons. The results indicate a statistically significant difference for the six metrics on the three datasets, except for the cases that are marked with (*) in Table 2.

### 3.3. Small Tumor Segmentation

The physical size for all images of the three datasets is not available. Therefore, the length of the longest axis of a tumor region in the original BUS image (non-resized) is used as a criterion for selecting small tumors, and the length threshold is set to 120 pixels. BUSIS, BUSI, and Dataset B contain 49, 151, and 76 small tumors, respectively. Figure 6 illustrates the FPR comparison between the overall and small tumor segmentation. All ten approaches have higher FPR for small tumors on BUSIS and Dataset B datasets. The FPR of AlexNet increases dramatically for small tumor segmentation. The ESTAN approach is superior in comparison to all nine approaches and achieves the lowest false positive for both overall and small tumor segmentation. Table 3 shows the results of all approaches on the three datasets using the selected seven quantitative metrics. The Enhanced Small Tumor-Aware Network outperforms all the other nine approaches for small tumor segmentation on the three datasets. AlexNet and SegNet obtain high TPRs, but at the cost of high FPRs.

### 3.4. Segmentation Tumors with Different Sizes

To demonstrate the effectiveness of the proposed ESTAN model, we split the BUSIS [20,26,38,39] dataset into four tumor-size groups. We chose the BUSIS dataset for the following reasons: (1) the images were collected from three hospitals using five ultrasound devices operated by different radiologists; (2) the ground truths of the BUSIS dataset have less bias because they were prepared by four experienced radiologists, where three radiologists generated tumor boundaries for each BUS image separately, and the fourth radiologist—a senior expert—judged and adjusted the majority voting results; and (3) all ten approaches achieved the best performance on the BUSIS dataset compared to BUSI and Dataset B. We chose the length of the longest axis of a tumor as a criterion for selecting tumor groups in the original BUS image. The first group contains 19 images with tumor sizes from 0 to 100 pixels, the second group has 30 images from 100 to 120 pixels, the third group consists of 81 images from 120 to 160 pixels, and the fourth group has 432 images from 160 to 533 pixels.

Table 4 lists the values of JI and FPR for the four tumor groups. AlexNet has poor performance for segmenting the small tumor group with JI of 0.57 and FPR of 0.97, while FPRs and JIs are improved dramatically in the other three groups. The results of segmenting tumors in both mid-size groups (100–120, and 120–160) are close to each other, e.g., CE-NET and SCAN have achieved the same JI with 0.81 and 0.80 in both groups, respectively. The results show that the tumor sizes between (0–100) are the most difficult cases, and all ten approaches cannot achieve as good performance as segmenting large tumors. On the other hand, for the fourth group containing large tumor sizes (>160 pixels) all approaches achieved better results than the other tumor groups. The proposed ESTAN achieved the highest JI and lowest FPR values in all tumor groups.

## 4. Discussions

The BUS images used in this work were obtained from different ultrasound devices with non-uniform settings, and vary in image resolution, tissue depth, and contrast. It is challenging to develop and train a robust deep model that performs consistently well on BUS images from different sources. As shown in Table 2, the performance of all approaches differs on images from different datasets. For instance, DenseU-Net achieved a JI of 0.74 on the BUSIS dataset, but its JI on Dataset B is only 0.60. To improve the robustness of deep learning models for BUS image segmentation, we recommend (a) involving large and diverse BUS datasets collected from different resources in model training, and (b) redesigning network architectures and training strategies to learn robust features from ultrasound images.

The preparation of a large, diverse, and annotated BUS image dataset could be time-consuming and prohibitively expensive. Therefore, in the short term, the more feasible strategy is to develop robust deep networks and training processes. The results in Table 2, Table 3 and Table 4 indicate that the proposed two-encoder network architecture and the row-column kernels could lead to more robust segmentation results. Another possible solution to this challenge is to develop image synthesis approaches that could generate realistic and diverse BUS images.

The strengths of this study include (a) utilizing the human breast anatomical layers to design convolution kernels, (b) using two encoders to learn features and three skip connections to transfer contextual information to the decoder to locate tumors more accurately, and (c) validating the efficacy and weakness of the proposed approach using extensive experiments on three publicly available datasets. Although ESTAN achieved remarkable results for segmenting tumors of various sizes on the three datasets, it failed to detect tumors in 29 extremely challenging cases, because these cases had high speckle noise, extremely low contrast, and no clear tumor boundaries. To extract features at different scales, ESTAN uses two encoders that require more parameters, memory, and computational resources. Therefore, optimizing ESTAN to eliminate unnecessary parameters and operations is significant, specifically for resource-constrained systems such as mobile devices. In the future, we will investigate long-range semantic information to improve the current approach.

## 5. Conclusions

In this work, we proposed the Enhanced Small Tumor-Aware Network (ESTAN) to improve the segmentation of small tumors in BUS images. The Enhanced Small Tumor-Aware Network is comprised of two encoder branches that extract and fuse image context information at different scales. The proposed ESTAN encoder applies row-column-wise kernels to adapt to breast anatomy. The decoder has three skip connections from the two encoders to fuse features. The new design enhances the performance by incorporating multi-scale features and breast anatomy into the encoder layers. The proposed architecture is sensitive to small breast tumors and identifies small tumors accurately. In addition, the approach achieves state-of-the-art performance in segmenting tumors of different sizes. We validated the proposed approach extensively using three datasets and compared it with the other nine breast tumor segmentation approaches. The results demonstrate that ESTAN achieves state-of-the-art performance on all datasets.

## Figures and Tables

**Figure 1 healthcare-10-02262-f001:**
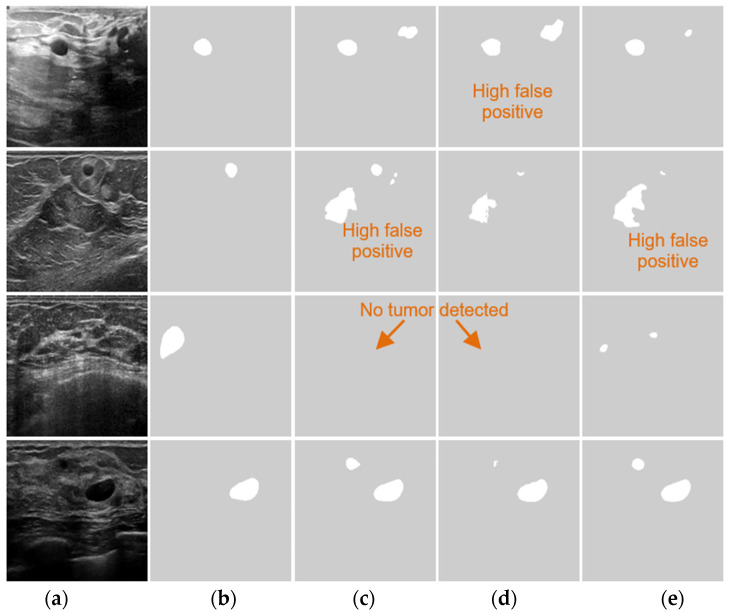
Performance of state-of-the-art approaches for segmenting breast tumors with different sizes. GT: Ground truth. (**a**) BUS Images; (**b**) GT; (**c**) DenseU-Net; (**d**) CE-Net; and (**e**) RDAU-Net. The arrows point to BUS images with no tumor detected.

**Figure 2 healthcare-10-02262-f002:**
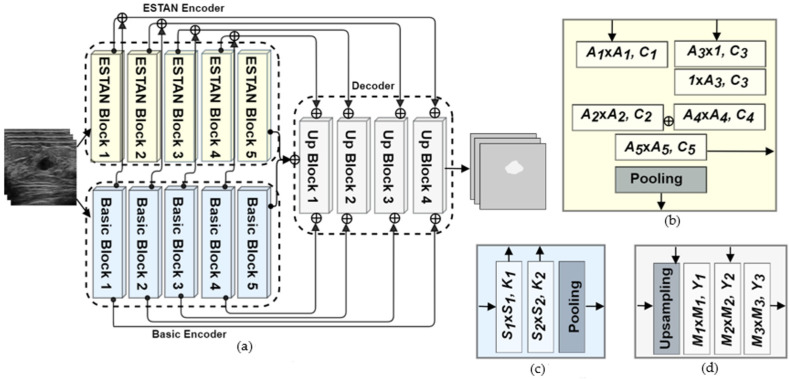
ESTAN architecture. (**a**) Overall architecture; (**b**) ESTAN block; (**c**) basic block; and (**d**) up block. ⊕ is the concatenation operator, *A_i_*, *S_i_*, *M_i_*, denote kernel sizes, and *C_i_*, *K_i_*, *Y_i_* define number of kernels.

**Figure 3 healthcare-10-02262-f003:**
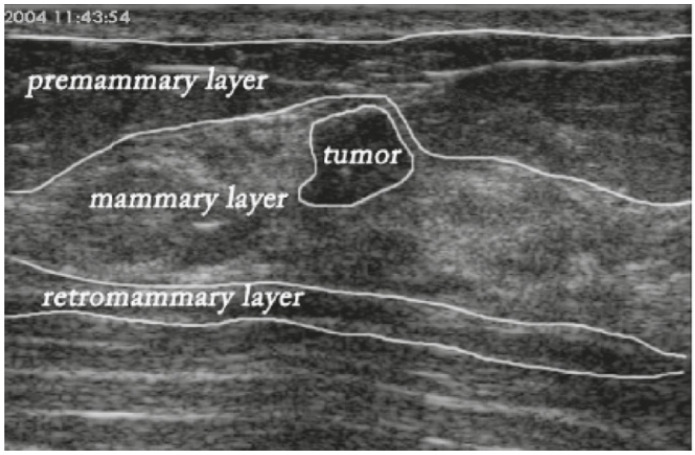
Major breast layers of a sample BUS image.

**Figure 4 healthcare-10-02262-f004:**
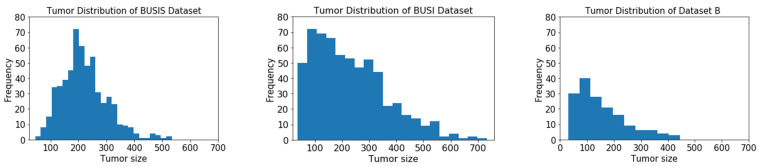
Histogram of tumor sizes (number of pixels).

**Figure 5 healthcare-10-02262-f005:**
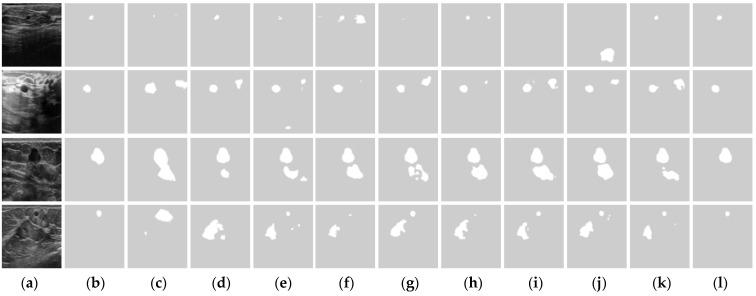
Tumor segmentation examples. (**a**) BUS Image, (**b**) ground truth, (**c**) AlexNet, (**d**) SegNet, (**e**) U-Net, (**f**) CE-Net, (**g**) MultiResUNet, (**h**) RDAU-Net, (**i**) SCAN, (**j**) DenseU-Net, (**k**) STAN, and (**l**) ESTAN.

**Figure 6 healthcare-10-02262-f006:**
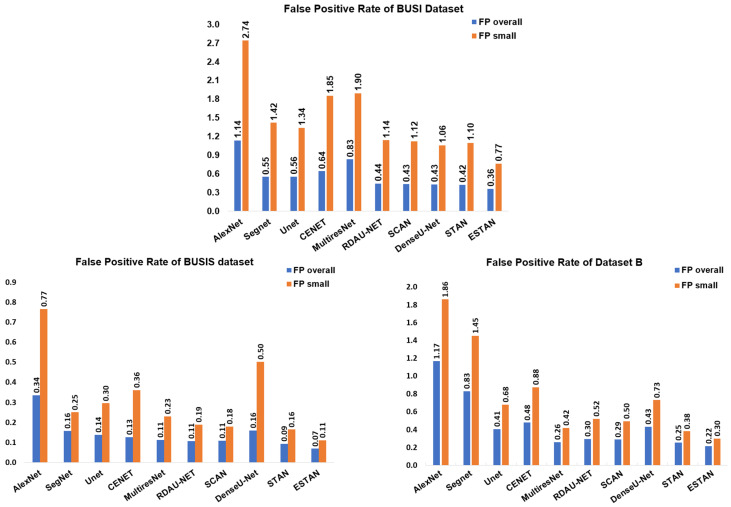
False positive rates of overall and small tumor segmentation on the three datasets.

**Table 2 healthcare-10-02262-t002:** Overall performance.

Datasets	Methods	TPR	FPR	JI	DSC	AER	HD	MAE
BUSIS [20,26,38,39]	AlexNet	**0.95**	0.34	0.74	0.84	0.39	25.1	7.1
	SegNet	0.94	0.16	0.82	0.90	0.22	21.7	4.5
	U-Net	0.92	0.14	0.83	0.90	0.22	26.8	4.9
	CE-Net	0.91	0.13	0.83	0.90	0.22	21.6	4.5
	MultiResUNet	0.93	0.11	0.84	0.91	0.19	18.8	4.1
	RDAU-NET	0.91	0.11	0.84	0.91	0.20	19.3	4.1
	SCAN	0.91	0.11	0.83	0.90	0.20	26.9	4.9
	DenseU-Net	0.91	0.16	0.81	0.88	0.25	25.3	5.5
	STAN	0.92	0.09	0.85	0.91	0.18	18.9	3.9
	ESTAN	0.91	**0.07**	**0.86**	**0.92**	**0.16**	**16.4**	**3.2**
Dataset B [3]	AlexNet	**0.87**	1.17	0.47	0.61	1.30	40.8	14.5
	SegNet	0.85	0.83	0.60	0.71	0.98	41.6	11.4
	U-Net	0.78	0.41	0.65	0.75	0.63	39.6	10.8
	CE-Net	0.74	0.48 *	0.61	0.72	0.74	40.1	10.5
	MultiResUNet	0.79	0.26	0.66	0.75	0.48	37.1	10.7
	RDAU-NET	0.78	0.30 *	0.67	0.77	0.52	32.4	8.3
	SCAN	0.75	0.29 *	0.65	0.74	0.54	43.7	9.9
	DenseU-Net	0.71	0.43	0.60	0.69	0.72	48.9	15.5
	STAN	0.80	0.27 *	0.70 *	0.78	0.47 *	35.5	9.7 *
	ESTAN	0.84	**0.22**	**0.74**	**0.82**	**0.38**	**25.5**	**7.0**
BUSI [40]	AlexNet	**0.87**	1.14	0.55	0.68	1.27	47.4	14.1
	SegNet	0.77	0.55	0.62	0.72	0.78	46.5	13.3
	U-Net	0.77	0.56	0.63	0.73	0.78	59.0	13.7
	CE-Net	0.77	0.64	0.64	0.73	0.88	43.9	12.4
	MultiResUNet	0.78	0.37	0.67	0.75	0.59	41.2	12.0
	RDAU-NET	0.80	0.42 *	0.68	0.76	0.62	39.2	12.0
	SCAN	0.73	0.43	0.63	0.72	0.70	47.0	13.8
	DenseU-Net	0.74	0.43	0.64	0.72	0.69	47.4	15.5
	STAN	0.76	0.42 *	0.66	0.75	0.66	46.5	12.1
	ESTAN	0.80	**0.36**	**0.70**	**0.78**	**0.56**	**34.8**	**9.9**

* refers to the statistically significant results. The bold results are the best performance according to a metric.

**Table 3 healthcare-10-02262-t003:** Performance of small tumor segmentation.

Datasets	Methods	TPR	FPR	JI	DSC	AER	HD	MAE
BUSIS [20,26,38,39]	AlexNet	**0.95**	0.77	0.60	0.73	0.82	26.3	9.6
	SegNet	0.92	0.25	0.75	0.84	0.33	22.4	6.2
	U-Net	0.92	0.30	0.76	0.84	0.38	44.2	8.3
	CE-Net	0.91	0.36	0.73	0.82	0.46	34.8	9.0
	MultiResUNet	0.91	0.23	0.77	0.84	0.33	27.7	8.5
	RDAU-NET	0.89	0.19	0.78	0.86	0.30	22.0	7.3
	SCAN	0.88	0.18	0.77	0.85	0.30	27.4	6.2
	DenseU-Net	0.90	0.50	0.72	0.81	0.60	34.5	8.2
	STAN	0.90	0.17	0.79	0.87	0.26	21.3	5.2
	ESTAN	0.90	**0.11**	**0.82**	**0.89**	**0.21**	**14.9**	**3.0**
Dataset B [3]	AlexNet	**0.87**	1.86	0.35	0.49	2.00	49.2	18.4
	SegNet	0.85	1.45	0.50	0.62	1.60	50.1	14.2
	U-Net	0.77	0.68	0.59	0.68	0.91	43.1	13.8
	CE-Net	0.72	0.88	0.53	0.63	1.15	50.0	14.4
	MultiResUNet	0.79	0.42	0.62	0.71	0.62	39.3	11.5
	RDAU-NET	0.78	0.52	0.62	0.71	0.73	34.1	8.8
	SCAN	0.75	0.50	0.61	0.70	0.74	48.7	11.2
	DenseU-Net	0.70	0.73	0.54	0.63	1.02	56.0	20.0
	STAN	0.81	0.40	0.67	0.76	0.59	35.9	11.1
	ESTAN	0.85	**0.30**	**0.72**	**0.80**	**0.44**	**21.5**	**6.3**
BUSI [40]	AlexNet	**0.94**	2.74	0.41	0.56	2.81	52.5	15.4
	SegNet	0.81	1.42	0.55	0.66	1.61	52.1	16.6
	U-Net	0.86	1.34	0.63	0.73	1.48	61.0	13.0
	CE-Net	0.83	1.86	0.59	0.69	2.03	50.9	13.3
	MultiResUNet	0.85	0.83	0.67	0.76	0.99	34.7	10.6
	RDAU-NET	0.87	0.99	0.68	0.77	1.13	33.9	9.9
	SCAN	0.80	1.13	0.63	0.73	1.33	42.4	12.5
	DenseU-Net	0.81	1.06	0.65	0.73	1.26	40.9	13.7
	STAN	0.86	1.10	0.67	0.76	1.25	49.2	11.3
	ESTAN	0.89	**0.77**	**0.72**	**0.81**	**0.88**	**24.2**	**6.1**

The bold results are the best performance according to a metric.

**Table 4 healthcare-10-02262-t004:** Performance of four tumor size groups of BUSIS dataset.

Tumor Size Groups	(0–100)		(100–120)		(120–160)		(>160)	
Number of Images	19		30		81		432	
	JI	FP	JI	FP	JI	FP	JI	FP
AlexNet	0.57	0.97	0.63	0.64	0.68	0.44	0.76	0.27
SegNet	0.71	0.28	0.77	0.23	0.79	0.21	0.83	0.14
U-Net	0.72	0.34	0.78	0.27	0.80	0.18	0.84	0.11
CE-Net	0.62	0.63	0.80	0.19	0.80	0.16	0.84	0.09
MultiResUNet	0.71	0.34	0.80	0.16	0.82	0.17	0.86	0.09
RDAU-NET	0.72	0.26	0.82	0.14	0.81	0.17	0.85	0.09
SCAN	0.71	0.24	0.81	0.14	0.81	0.16	0.80	0.09
DenseU-Net	0.67	0.77	0.75	0.34	0.78	0.21	0.83	0.11
STAN	0.76	0.25	0.81	0.11	0.83	0.12	0.86	0.08
ESTAN	0.79	0.15	0.83	0.09	0.85	0.10	0.87	0.06

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
