# Peer review of "ESTAN: Enhanced Small Tumor-Aware Network for Breast Ultrasound Image Segmentation"

_healthcare, 2022, doi:10.3390/healthcare10112262_

Round 1

Reviewer 1 Report

The abstract needs modification with quantification of wide results. Section 2 needs enhancement. In the results MCC, Kappa and Error rate to be calculated and analyzed. Discussion  section needs strengthen. conclusion needs modification. The objectives and contribution has to be clearly indicated and discussed. Some statistical tests may be performed on the data sets.

Author Response

We appreciate the valueable comments. The manuscript has been revised accordingly and carefully.

(1) The abstract needs modification with quantification of wide results.

Response:  We have modified the abstract. (Page 1)

We compare ESTAN and nine state-of-the-art approaches using seven quantitative metrics on three public breast ultrasound datasets, i.e., BUSIS, Dataset B, and BUSI. The results demonstrate that the proposed approach achieves the best overall performance and outperforms all other approaches on small tumor segmentation. Specifically, the Dice similarity coefficient (DSC) of ESTAN on the three datasets is 0.92, 0.82, and 0.78, respectively; and the DSC of ESTAN on the three datasets of small tumors is 0.89, 0.80, and 0.81, respectively.

(2) Section 2 needs enhancement.

Response: We have enhanced Section 2. Specifically, we (1) improved the grammar, (2) corrected several references, and (3) rephrased several sentences to improve clarity. For instance, the first paragraph of Section 2 is revised as below.

“… To alleviate these challenges, we propose ESTAN to extract and fuse image context information at different scales. ESTAN constructs feature maps using both square and large row-column-wise kernels. These feature maps extract multi-scale context information and preserve fine-grained tumor location information. Therefore, the new design enables ESTAN to segment breast tumors of different sizes accurately, and is especially effective in segmenting small tumors. The overall architecture of the proposed approach is shown in Figure 2.”

(3) In the results MCC, Kappa and Error rate to be calculated and analyzed.

Response: Thanks for the suggestion. In the evaluation, we used seven metrics, true positive rate (TPR), false positive rate (FPR), Jaccard index (JI), Dice similarity coefficient (DSC), area error rate (AER), Hausdorff distance (HD), and mean absolute error (MAE), which were recommended in [35] to evaluate the segmentation approaches for breast ultrasound images. The first five metrics measure the agreement of the generated tumor region (TP + FP) with the true tumor region (TP + FN), and the last two measure the agreement between the generated tumor contour and the true tumor contour. Both Kappa and MCC are robust measures that take into account true and false positives and negatives. They are not used because the positives (tumor regions) and negatives (non-tumor regions) are not equally important in this application, and all recommended metrics focus on the tumor regions.

(4) Discussion section needs strengthen.

Response: We have improved the Discussion Section. (Page 12)

“BUS images used in this work were obtained from different ultrasound devices with non-uniform settings, and vary in image resolutions, tissue depths, and contrast. It is challenging to develop and train a robust deep model that performs consistently well on BUS images from different sources. As shown in Table 2, the performance of all approaches differs on images from different datasets. For instance, DenseU-Net achieved a JI of 0.74 on the BUSIS dataset, but its JI on the Dataset B is only 0.60. To improve the robustness of deep learning models for BUS image segmentation, we recommend (a) involving large and diverse BUS datasets collected from different resources in model training, and (b) redesigning network architectures and training strategies [MICCAI 23] to learn robust features from ultrasound images.

The preparation of a large, diverse, and annotated BUS image dataset could be time-consuming and prohibitively expensive. Therefore, in the short term, the more feasible strategy is to develop robust deep networks and training processes. The results in Tables 2, 3, and 4 indicate that the proposed two-encoder network architecture and the row-column kernel could lead to more robust segmentation results. Another possible solution to this challenge is to develop image synthesis approaches that could generate realistic and di-verse BUS images.

The strengths of this study include (a) utilizing the human breast anatomical layers to design convolution kernels, (b) using two encoders to learn features and three skip connections to transfer contextual information to the decoder to locate tumors more accurately, and (c) validating the efficacy and weakness of the proposed approach using extensive experiments on three publicly available datasets. Although ESTAN achieved remarkable results for segmenting tumors of various sizes on the three datasets, it failed to detect tumors in 29 extremely challenging cases because these cases have high speckle noise, extremely low contrast, and no clear tumor boundaries. To extract features at different scales, ESTAN uses two encoders which require more parameters, memory, and computational resources. Therefore, optimizing ESTAN to eliminate unnecessary parameters and operations is significant, specifically for resource-constrained systems such as mobile devices. In the future, we will investigate the long-range semantic information to improve the current approach.”

(5) Conclusion needs modification.

Response: We have improved the conclusion. (Pages 12 and 13)

“In this work, we proposed the Enhanced Small Tumor-Aware Network (ESTAN) to improve the segmentation of small tumors in BUS images. ESTSAN is comprised of two en-coder branches that extract and fuse image context information at different scales. The proposed ESTAN encoder applies row-column-wise kernels to adapt to the breast anatomy. The decoder has three skip connections from the two encoders to fuse features. The new design enhances the performance by incorporating multi-scale features and breast anatomy into the encoder layers. The proposed architecture is sensitive to small breast tumors and identifies small tumors accurately. In addition, the approach achieves state-of-the-art performance in segmenting tumors of different sizes. We validate the proposed approach extensively using three datasets and compare it with the other nine breast tumor segmentation approaches. The results demonstrate that ESTAN achieves state-of-the-art performance on all datasets.”

(6) The objectives and contribution has to be clearly indicated and discussed.

Response: Done. The objectives have been discussed in the third paragraph of Page 4; and we have summarized and discussed the contributions in the Discussion and Conclusion Sections.

(7) Some statistical tests may be performed on the data sets.

Response: The Wilcoxon signed-rank test was employed to investigate the statistical significance of all approaches. (Page 8)

Reviewer 2 Report

In this paper, a novel deep neural network architecture is proposed to accurately and robustly segment breast tumors. The overall writing is good and the work is interesting. Some questions are better answered before the manuscript is accepted for publication.

(1) The authors should consider more recent research done in the field of their study. Such as:

"Breast cancer detection based on thermographic images using machine learning and deep learning algorithms"

"Breast cancer detection using deep learning: Datasets, methods, and challenges ahead"

"COVID-19 detection using chest X-ray images based on a developed deep neural network"

(2) State the weaknesses and strengths of your model.

(3) The Limitations of the proposed study need to be discussed before the conclusion.

(4) Dataset seems to be imbalanced. How samples have been balanced for different classes?

(5) The proposed method is sensitive to the values of its main hyper-parameters. How did the authors tune the hyper-parameters?

Author Response

We appreciate the valueable comments. The manuscript has been revised accordingly and carefully.

In this paper, a novel deep neural network architecture is proposed to accurately and robustly segment breast tumors. The overall writing is good and the work is interesting. Some questions are better answered before the manuscript is accepted for publication.

(1) The authors should consider more recent research done in the field of their study. Such as:

"Breast cancer detection based on thermographic images using machine learning and deep learning algorithms"

"Breast cancer detection using deep learning: Datasets, methods, and challenges ahead"

"COVID-19 detection using chest X-ray images based on a developed deep neural network"

 Response: Thanks for suggesting the references. In this work, we are focusing on breast tumor segmentation in ultrasound images, and we have added three recent papers ([34], [45], and [46]) in this area.

(2) State the weaknesses and strengths of your model.

 Response: We have discussed the weakness and strengths of the proposed mode in the third paragraph of the Discussion Section. (Page 12)

“The strengths of this study include (a) utilizing the human breast anatomical layers to design convolution kernels, (b) using two encoders to learn features and three skip connections to transfer contextual information to the decoder to locate tumors more accurately, and (c) validating the efficacy and weakness of the proposed approach using extensive experiments on three publicly available datasets. Although ESTAN achieved remarkable results for segmenting tumors of various sizes on the three datasets, it failed to detect tumors in 29 extremely challenging cases because these cases have high speckle noise, extremely low contrast, and no clear tumor boundaries. To extract features at different scales, ESTAN uses two encoders which require more parameters, memory, and computational resources. Therefore, optimizing ESTAN to eliminate unnecessary parameters and operations is significant, specifically for resource-constrained systems such as mobile devices. In the future, we will investigate the long-range semantic information to improve the current approach.

(3) The limitations of the proposed study need to be discussed before the conclusion.

Response: We have discussed the limitation of ESTAN in the Discussion Section.

“Although ESTAN achieved remarkable results for segmenting tumors of various sizes on the three datasets, it failed to detect tumors in 29 extremely challenging cases because these cases have high speckle noise, extremely low contrast, and no clear tumor boundaries. To extract features at different scales, ESTAN uses two encoders which require more parameters, memory, and computational resources. ”

(4) Dataset seems to be imbalanced. How samples have been balanced for different classes?

Response: It is true that the number of background pixels is much larger than the number of tumor pixels. We need to balance different classes if the cross-entropy loss is applied. In this work, the Dice loss function was applied; and it only considers the agreement between the true tumor regions and predicted tumor regions. Therefore, there is no need to balance the two classes for the Dice loss function..

(5) The proposed method is sensitive to the values of its main hyper-parameters. How did the authors tune the hyper-parameters?

Response: We have searched the best hyper-parameters by experiments. For example, to find the row-column-wise kernels, we have conducted experiments to compare different combinations of kernel sizes for each block.

Author Response

We appreciate the valueable comments. The manuscript has been revised accordingly and carefully.

This manuscript has explored the tumor segmentation problem by introducing ESTAN to specifically improve small tumor segmentation. The paper can be further improved by addressing the following concerns:

(1) Why False Positive Rate is highlighted as the primary performance measure in the abstract? Is it considered the most important measure compared to the others?

Response: Figure 1 shows that one of the major issues of current state-of-the-art approaches is the high false positive rate (FRP). Therefore, reducing FPR is one of the major goals of this work. In addition, we have modified the Abstract and highlighted more comprehensive metric, e.g., the DSC.. (Page 1)

“Specifically, the Dice similarity coefficient (DSC) of ES-TAN on the three datasets is 0.92, 0.82, and 0.78, respectively; and the DSC of ESTAN on the three datasets of small tumors is 0.89, 0.80, and 0.81, respectively.”

(2) “To the best of our knowledge, STAN [3] was the firsit deep learning architecture designed specifically for small tumor segmentation”-> The suggested paper does not refer to any deep learning method, instead they use conventional SVM? Reference no. 3 was published in 2005.

Response: We have modified the reference as below. (Page 4)

” …To the best of our knowledge, STAN [15] was the first deep learning architecture designed specifically for small tumor segmentation. Three skip connections and two encoders were employed to extract multi-scale contextual information from different layers of the contracting part. STAN outperformed other deep learning approaches for segmenting small tumors in BUS images.”

“15. B. Shareef, M. Xian, and A. Vakanski, “STAN : Small Tumor-Aware Network for Breast Ultrasound Image Segmentation,” IEEE 17th Int. Symp. Biomed. Imaging (ISBI 2020), 2020.”

(3) How the proposed row-column-wise kernels, A3x1 followed by 1xA3 is a better feature extractor than A3xA3? What is the assumption as generally, a piece of better neighborhood information can be extracted from A3xA3 rather than A3x1 followed by 1xA3? Furthermore, a small tumor still looks like a round shape, that has an almost equal length in height and width?

Response: The assumption is that the row-column-wise kernels (A3x1 and 1xA3) model global context information more effectively and could lower the false positive rate (FPR). The proposed ESTAN encoder (1) avoids excessive pooling operations and preserves small tumor regions, and (2) distinguishes small tumors from normal tissues by using more effective global context information that aligns with breast anatomy.

(4) The arrows in Figure 2 can be improved as some arrows are overlapping with the bounding boxes.

Response: We have fixed it.

(5) How the kernels are chosen in “applied kernels of size 1x1, 3x3, and 5x5? Is it through experimentation?

Response: The 1x1, 3x3, and 5x5 kernels were proposed in STAN [15], and were chosen by experiments..

 (6) “To overcome the singularity issues during the training, we have introduced three skip connections" -> the application of skip connection is more famous to overcome the vanishing gradient problem for a deep network? It is rare for a skip connection to be associated with a singularity issue?

Response: We have revised the second paragraph of Section 2.3. (Page 7)

“ We have introduced skipping connections to copy feature maps at different scales from two encoders to the decoder. The first two skip connections combine the result of f_(S_1,K_1 ) in the basic encoder block and the result of f_(A_5,C_5 ) in the ESTAN encoder block and are concatenated to the upsampling layer. The third skip connection concatenates the results of f_(S_2,K_2 ) and f_(M_2,Y_2 ). The output layer utilizes a 1×1 convolution followed by a sigmoid activation to predict the final results. Figure 2(d) illustrates the decoder block. ”

Round 2

Reviewer 1 Report

The paper may be accepted.

Reviewer 3 Report

The authors have addressed all my comments.